# Procyanidins Alleviated Cerebral Ischemia/Reperfusion Injury by Inhibiting Ferroptosis via the Nrf2/HO-1 Signaling Pathway

**DOI:** 10.3390/molecules28083582

**Published:** 2023-04-20

**Authors:** Lei Chen, Jie Huang, Zi-Meng Yao, Xiao-Rong Sun, Xu-Hui Tong, Miao Hu, Ying Zhang, Shu-Ying Dong

**Affiliations:** 1Department of Pharmacology, School of Pharmacy, Bengbu Medical College, Bengbu 233030, China20201007007@stu.bbmc.edu.cn (J.H.);; 2Anhui Engineering Technology Research Center of Biochemical Pharmaceutical, Bengbu 233030, China; 3Key Laboratory of Cardiovascular and Cerebrovascular Diseases, Bengbu Medical College, Bengbu 233030, China

**Keywords:** procyanidins, cerebral, ischemia/reperfusion, ferroptosis, Nrf2/HO−1pathway, mice

## Abstract

Procyanidins (PCs), which are organic antioxidants, suppress oxidative stress, exhibit anti−apoptotic properties, and chelate metal ions. The potential defense mechanism of PCs against cerebral ischemia/reperfusion injury (CIRI) was investigated in this study. Pre−administration for 7 days of a PC enhanced nerve function and decreased cerebellar infarct volume in a mouse middle cerebral artery embolization paradigm. In addition, mitochondrial ferroptosis was enhanced, exhibited by mitochondrial shrinkage and roundness, increased membrane density, and reduced or absent ridges. The level of Fe^2+^ and lipid peroxidation that cause ferroptosis was significantly reduced by PC administration. According to the Western blot findings, PCs altered the expression of proteins associated with ferroptosis, promoting the expression of GPX4 and SLC7A11 while reducing the expression of TFR1, hence inhibiting ferroptosis. Moreover, the treatment of PCs markedly elevated the expression of HO−1 and Nuclear−Nrf2. The PCs’ ability to prevent ferroptosis due to CIRI was decreased by the Nrf2 inhibitor ML385. Our findings showed that the protective effect of PCs may be achieved via activation of the Nrf2/HO-1 pathway and inhibiting ferroptosis. This study provides a new perspective on the treatment of CIRI with PCs.

## 1. Introduction

Stroke is one of the leading causes of death and disability worldwide, with ischemic stroke accounting for 80% of cases [1]. The main treatment is the restoration of blood supply as soon as possible [2]. Unfortunately, blood reperfusion, the increase in reactive oxygen species (ROS), and the antioxidant defense capability of the system are weakened, followed by an excessive production of oxygen free radicals and antioxidant enzyme inactivation. Consequent adverse events include protein and lipid peroxidation [3]. The brain is rich in polyunsaturated fatty acids (PUFAs) and iron and is easily damaged by lipid peroxidation [4]. Lipid peroxidation and disrupted iron metabolism have recently been identified as key factors in a new form of cell death, ferroptosis, the occurrence of which further aggravates cerebral I/R injury [5].

Procyanidins (PCs) are polyphenolic antioxidants widely found in plants that can effectively remove free radicals and protect lipids from peroxidation damage [6]. They are also powerful metal-chelating agents [7]. Studies have shown that PCs can alleviate cerebral I/R injury [8]. Nonetheless, few studies have examined the mechanisms that underlie the protective effect of PCs against CIRI. Nrf2 is an oxidative stress sensor and a key transcription factor that protects cells against oxidative damage caused by exogenous stimulation. Under physiological conditions, Nrf2 is mainly present in the cytoplasm. Nonetheless, when the ROS levels of antioxidant response elements (ARE) are elevated, Nrf2 is transferred to the nucleus to bind to ARE and regulate many antioxidant coding genes, especially HO−1 expression [9,10]. The Nrf2/HO−1 pathway is heavily involved in protection against cerebral I/R injury [10,11]. It has been found that Nrf2 inhibits ferroptosis induced by acute lung I/R injury by regulating SLC7A11 and HO−1 [12]. Nonetheless, whether the protective effect of PC is related to ferroptosis and its relationship with the Nrf2/HO−1 pathway has not been reported.

This study explored the role and mechanism of PCs in a mouse model of CIRI, to aid in the search for pharmacological agents that can prevent CIRI.

## 2. Results

### 2.1. PCs Afford Protection against CIRI

In this study, different doses of PC (25, 50, or 100 mg·kg^−1^) were intragastrically administered to determine its role in CIRI. Chemical structure of procyanidins and the experimental plan shown in Figure 1. All mice in the I/R group and administration groups were subjected to 1 h ischemia followed by reperfusion for 24 h. As shown in Figure 2A, mice in the sham group showed regular behavior. Those in the I/R group had significant neurological deficits, evidenced by the high neurological score. PC supplementation markedly improved the neurological deficits. As shown in Figure 2B,C, TTC staining confirmed that the infarct volume was significantly reduced in the PC−treated I/R group (25 mg·kg^−1^, 50 mg·kg^−1^, and 100 mg·kg^−1^) in a dose−dependent manner. Compared with the I/R group, there was no significant difference between the PC−treated I/R group (100 mg·kg^−1^) and the edaravone group.

In addition, HE staining and Nissl staining were used to evaluate the protective effect of PCs on CIRI. As shown in Figure 2D, HE staining revealed that cortical neurons in the sham group had regular morphology, clear nucleoli, abundant cytoplasm, and close arrangement. In the I/R group, cerebral cortical neuron damage was obvious, evidenced by swelling, chromatin condensation, disordered neuron arrangement, and reticular lesions. As shown in Figure 2E, Nissl staining revealed severe loss of Nissl bodies following I/R injury. Compared with the I/R group, the PC treatment group (100 mg·kg^−1^) had an increased number of Nissl bodies. Collectively, these results show that PCs could effectively ameliorate CIRI.

### 2.2. PCs Inhibit Ferroptosis in Mice following CIRI

Transmission electron microscopy (TEM) was used to observe the morphology of mitochondria in tissue on the infarcted side following CIRI. As shown in Figure 3A, tissue on the infarcted side showed typical changes of ferroptosis, mitochondrial contraction, increased membrane density, and crest reduction. This change was reversed by PC administration. To determine whether the protective effect of PCs on CIRI was related to ferroptosis, Western blot was used to detect the expression of ferroptosis-related proteins including GPX4, SLC7A11, and TFR1. GPX4 and SLC7A11 are the main anti-lipid peroxidation proteins, and the TFR1 protein is closely related to the Fenton reaction. As shown in Figure 3B−E, the expression of GPX4 and SLC7A11 in the I/R group was lower than that of the sham group, while the expression of TFR1 was higher. Nonetheless, the expression of GPX4 and SLC7A11 was up−regulated in the I/R + PC group compared with the I/R group, while TFR1 was down-regulated. We also determined the level of Fe^2+^, MDA, and GSH. As shown in Figure 3F,G, the level of Fe^2+^ and MDA increased in the I/R group but decreased following PC administration. The level of GSH decreased in the I/R group but increased after PC administration. These results suggest that PC administration improved ferroptosis in mice with cerebral I/R.

### 2.3. PCs Activate the Nrf2/HO-1 Pathway

Western blot was used to measure the expression of Nuclear−Nrf2 and HO−1 proteins to further explore the mechanism of ferroptosis activated by PCs. As shown in Figure 4A−D, CIRI significantly increased the expression of both Nuclear-Nrf2 but decreased that of HO-1. Nonetheless, PCs significantly increased the expression of Nuclear−Nrf2 and HO-1. These results suggest that PC administration could ameliorate CIRI and may be related to the Nrf2/HO-1 pathway.

### 2.4. ML385 Reverses the Neuroprotective Effect of PCs

To explore the relationship between Nrf2 and the protective effect of PCs on CIRI, the Nrf2 inhibitor ML385 was applied since the protective effect of PCs was likely conferred via the Nrf2 pathway. As shown in Figure 5A−E, ML385 significantly inhibited the protective effect of PCs against CIRI. These results suggest that PCs play a neuroprotective role by activating the Nrf2 pathway.

### 2.5. ML385 Reverses the Anti-Ferroptotic Effects of PCs in Mice following CIRI

To further explore the relationship between Nrf2 and the effect of PCs on ferroptosis in mice with CIRI, we examined the extent of ferroptosis after administration of ML385. As shown in Figure 6A, compared with the PC_100_ group, the mitochondria of the ML385 group exhibited changes typical of ferroptosis. As shown in Figure 6B–E, ML385 reversed the expression of ferroptosis-related proteins GPX4, SLC7A11, and TFR1. Moreover, ML385 reversed the level of ferroptosis-related indicators Fe^2+^, MDA, and GSH (Figure 6F,G). These results suggest that PCs play a neuroprotective role by activating the Nrf2 pathway to inhibit ferroptosis. 

### 2.6. ML385 Inhibits the Nrf2/HO-1 Pathway

To determine whether the Nrf2/HO−1 signal pathway plays an essential role in cerebral I/R−induced ferroptosis, the expression of Nuclear-Nrf2 and HO−1 was evaluated following inhibition of the Nrf2 pathway. As shown in Figure 7A−D, ML385 significantly decreased the expression of Nuclear−Nrf2 and HO-1. These results suggest that the protective effect of PCs against CIRI may be achieved by alleviating ferroptosis via the Nrf2/HO−1 pathway.

## 3. Discussion

In this study, a mouse model of middle artery embolization was established. A course of PC was administered by gavage 1 h before ischemia and for 7 days thereafter with a consequent reduction in infarct size and improved neurobehavioral score. We also explored the protective mechanism of PCs against CIRI. We demonstrated that: (1) PCs improved neurobehavioral scores and reduced infarct volume in mice with CIRI; (2) PCs had a good inhibitory effect on CIRI-induced ferroptosis; and (3) Further mechanistic study suggested that the inhibitory effect of PCs on ferroptosis may be related to the Nrf2/HO−1 pathway. 

CIRI is a complex pathological process that involves multiple mechanisms. The clinical treatment of cerebral ischemia is mainly thrombolytic therapy to restore blood recanalization. Nonetheless, blood recanalization can further aggravate brain injury [13]. This is clearly an urgent clinical problem that warrants further exploration to understand the mechanism of CIRI and develop new drugs that can prevent CIRI consequent to blood recanalization following thrombolysis.

Recent studies have found that ferroptosis is an important factor in CIRI after stroke [14,15]. Ferroptosis is a non-apoptotic regulatory cell death induced by iron-dependent excessive accumulation of phospholipid peroxides [16]. Mitochondria shrink and become round and membrane density increases and crest decreases which are typical changes associated with ferroptosis [17], all evident in the I/R group and alleviated by PC administration. The phospholipid peroxidation of cell membrane lipid bilayers is the key driver of ferroptosis [18]. When the endogenous antioxidant state of the cell is damaged, lipid ROS accumulation and destruction of membrane structure will trigger ferroptosis. Glutathione peroxidase 4 (glutathioneperoxidase4, GPX4) is produced on the cell membrane and can directly reduce the level of intracellular lipid peroxides [19]. The decreased GPX4 activity is insufficient to inhibit the production of intracellular ROS, resulting in the accumulation of lipid peroxides and the production of lethal ROS and consequent ferroptosis [20]. GPX4 is regulated by upstream SLC7A11, and glutathione (GSH) is a key substrate for GPX4 synthesis. SLC7A11−GSH−GPX4 is the classic ferroptosis route. As characteristic proteins for detecting ferroptosis, GPX4 and SLC7A11 have been shown to decrease significantly when ferroptosis occurs in nerve cells [21]. Our study determined that GPX4 and SLC7A11 expression decreased following CIRI. Meanwhile, the GSH level decreased after the I/R injury but increased after PC administration. Another factor in ferroptosis is iron, a cofactor in a series of biochemical processes that include oxygen storage, oxidative phosphorylation, and enzymatic reactions required for cell proliferation [22]. Iron exists mainly in the form of Fe^3+^ bound to transferrin in vivo and enters the cell through membrane transferrin receptor 1 (transferrin receptor 1, TFR1) binding endocytosis [23]. In vivo, Fe^3+^ can be converted to Fe^2+^ through REDOX reaction, and unstable Fe^2+^ can mediate the Fenton reaction to catalyze the formation of lipid reactive oxygen species, resulting in ferroptosis [24]. MDA is a product of lipid peroxidation and its level is an index of the degree of lipid peroxidation in the cell membrane [25]. Thus, changes in MDA and Fe^2+^ levels indicate the severity of ferroptosis. In our study, MDA and Fe^2+^ level was increased following middle cerebral artery occlusion/reperfusion (MCAO/R) in mice, accompanied by increased expression of the TFR1 protein. Nonetheless, the administration of PCs reduced the elevation of aggravation markers of ferroptosis. These results suggest that the neuroprotective effect of PCs in CIRI may be through the inhibition of ferroptosis.

Polyphenols have a wide range of pharmacological applications in CIRI and have attracted much attention. Procyanidins (PCs) are natural biological polyphenols with the effects of anti-oxidation, anti-apoptosis, and the inhibition of inflammation. They have potential therapeutic benefits in various diseases such as cerebral ischemia, Alzheimer’s disease, and cerebrovascular diseases [8,26,27]. The key role of PCs in the prevention of cardiovascular and cerebrovascular diseases is their contribution to the stabilization of the antioxidant system [26]. The results of our present study revealed that PCs dramatically protected nerve function and reduced cerebral infarct volume and pathological lesions in vivo, similar to a previous report [8]. This indicated that PC administration significantly preserved nerve function while reducing brain infarct extent and pathological lesions in vivo. In this work, we determined that PCs could significantly increase neurobehavioral test results and reduce the size of mouse cerebellar infarcts. Nonetheless, there was no statistically significant difference between the protective impact of PCs on CIRI and that of the edaravone group in the positive control group, indicating that PCs had a strong therapeutic effect on CIRI. Further study is required to determine how PCs protect mice against CIRI. Following I/R injury, TEM revealed that mitochondria displayed the expected changes associated with ferroptosis (rounder mitochondria, increased membrane density, and reduced ridge). Yet, following treatment with PCs, the mitochondrial ferroptotic properties abated substantially. In a previous study, an extract high in PCs prevented DNA damage during the Fenton reaction, and this protection was linked to the chelation of Fe^2+^ [28]. In our study, the administration of PCs greatly decreased the level of MDA, a byproduct of lipid peroxidation, and Fe^2+^ involved in the Fenton reaction. The Fenton reaction and anti-lipid peroxidation, which are essential to ferroptosis, are strongly related to PCs. These findings suggest that ferroptosis suppression may play a role in the protection afforded by PCs against CIRI. We also examined the expression of the proteins GSH, SLC7A11, and GPX4 which are associated with ferroptosis to further confirm the connection between PCs and ferroptosis. In mice with CIRI, PCs reduced ferroptosis and, intriguingly, activated the Nuclear-Nrf2 protein. Whether this increase in Nuclear−Nrf2 is related to the inhibitory effect of PCs on ferroptosis requires further study. Under oxidative stress, the Nrf2 protein escapes degradation and translocates to the nucleus, inducing the expression of heme oxygenase−1 (HO−1) and other genes, contributing to the antioxidant response [29]. Wang et al. [30] found that the expression of the Nuclear−Nrf2 protein by myocardial H9C2 cells was significantly increased following hypoxia/reoxygenation injury, and dexmedetomidine administration further increased its expression in cardiomyocytes. The change in Nuclear-Nrf2 expression was consistent with the results we observed in CIRI. Therefore, we further explored the correlation of Nrf2 with the protective effect of PC in CIRI.

Nuclear factor E2 related factor 2 (Nrf2) is a transcription factor. The target gene of Nrf2 mediates the regulation of many cellular functions including the endogenous antioxidant system, iron metabolism, and lipid metabolism [31]. Nrf2 plays a neuroprotective role in a variety of diseases such as stroke, Alzheimer’s disease, and Parkinson’s disease [32,33,34]. It has been reported that ferroptosis−related genes regulated by Nrf2 transcription include GSH, SLC7A11, and GPX4. Nrf2 and its target genes play an important role in maintaining the balance between cells and organelles. Inhibition of Nrf2 by ML385 can aggravate CIRI [35]. Importantly, Nrf2 is particularly involved in inducing the expression of heme oxygenase−1 (HO−1), an enzyme that degrades heme to ferrous, carbon monoxide, and biliverdin, and then reduces biliverdin reductase to bilirubin [36]. Recently, it has been found that up-regulation of HO−1 promotes heme degradation and ferritin synthesis to change the iron distribution in cells, and enhanced expression of HO−1 can induce ferroptosis by promoting iron accumulation and ROS production [37]. It has been confirmed that Nrf2 can up-regulate the expression of HO−1 and increase the expression of ferroptosis-related proteins downstream, thus increasing the sensitivity of cells to ferroptosis [33]. This suggests that the Nrf2/HO-1 pathway plays an important role in the resistance to ferroptosis. Our study found that PCs further activated Nuclear−Nrf2, and the Nrf2 downstream protein HO−1 also increased. To further elucidate the protective mechanism of PCs against CIRI, we applied Nrf2 inhibitor ML385. The results suggested that ML385 reversed the protective effect of PCs on CIRI, and down−regulated the expression of Nuclear-Nrf2, HO−1, and ferroptosis-related proteins. As illustrated in Figure 8, the results of this study suggest that the protective effect of PCs on CIRI may be achieved through the alleviation of ferroptosis via the Nrf2/HO−1 pathway.

CIRI leads to lipid peroxidation and Fe^2+^ dysregulation that decreases intracellular GPX4 levels. Fe^2+^ increase and lipid peroxidation lead to ferroptosis through the Fenton reaction. Procyanidins alleviated CIRI by activating the Nrf2/HO−1 pathway to inhibit intracellular Fe^2+^ level and lipid peroxidation, thereby inhibiting ferroptosis.

## 4. Materials and Methods

### 4.1. Experimental Animals and Groups

Male ICR mice (14 weeks of age) were purchased from the Experimental Animal Center of Anhui Medical University, Anhui, China. All mice were maintained under a 12 h/12 h light/dark cycle, and all animal experiments were approved by the Ethics Committee Experimental Animal Ethics Branch of Bengbu Medical College. For in vivo experiments on neuroprotective effects, all mice were divided into five groups: sham, I/R, I/R + PC_25_ (25 mg/kg), I/R + PC_50_ (50 mg/kg), I/R + PC_100_ (100 mg/kg), and I/R + Edaravone (30 mg/kg). For in vitro experiments to determine the mechanism of action of PCs, mice were divided into four groups: sham, I/R, I/R + PC_100_ (100 mg/kg), and I/R + PC_100_ + ML385 (30 mg/kg, Nrf2 inhibitor).

### 4.2. Drugs and Reagents

Drugs and reagents were obtained as follows: Procyanidins (C30H26O13, molecular weight = 594.52, purity > 98%) were purchased from Solarbio and were ultrasonically soluble in saline. The administration was for 7 days via gavage in the PC group and by intraperitoneal injection in the edaravone group. Nrf2 inhibitor (ML385, 30 mg/kg, MCE, Plainsboro, NJ USA) was administered via intraperitoneal injection prior to the establishment of the MCAO/R model.

### 4.3. Cerebral Ischemia/Reperfusion (I/R) Injury Model

CIRI was induced by MCAO/R as previously described [38]. Briefly, mice were anesthetized with pentobarbital sodium (40 mg/kg, intraperitoneal injection, Macklin, Shanghai, China) and the external carotid artery (ECA) and internal carotid artery (ICA) exposed through a midline neck incision under a surgical microscope. A nylon filament was inserted into the ICA via the ECA and slowly advanced to the middle cerebral artery (MCA) until mild resistance was felt. After occluding the right MCA for 1 h, the nylon filament was removed to restore blood flow. Mice in the sham group underwent the same procedure with the exception of MCA occlusion.

### 4.4. Neurological Score Evaluation

The behavioral deficits of mice were evaluated by the Longa score [39] with a range of 0 to 4 points: 0 points: no obvious neurological dysfunction; 1 point: weakness of and an inability to fully straighten left forelimb; 2 points: rotation of trunk to the opposite side when walking; 3 points: tipping of trunk on the opposite side while walking; and 4 points: unable to walk or comatose.

### 4.5. Cerebral Infarction Volume Measurement

Cerebral infarction volume was measured by staining with 1% 2,3,5-triphenyl tetrazolium chloride (TTC) (Saiguo biotech Co. Ltd., Guangzhou, China). After 24 h of reperfusion, the whole brain was removed quickly by decapitation, then frozen at −20 °C for 10 min. All brains were cut into four pieces along the coronal plane with an average thickness of 2 mm. Brain slices were stained with 1% TTC solution in the dark for 20 min at 37 °C, then fixed with 4% paraformaldehyde for 4 h and photographed. The percentage of cerebral infarct volume was calculated.

### 4.6. Hematoxylin–Eosin (HE) Staining

The brains were removed 24 h after reperfusion and fixed in 4% paraformaldehyde. Brain tissue was cut into 10 μm slices after dehydration and paraffin embedding. After dewaxing, the samples were washed with graded ethanol series (100%, 95%, 80%, and 75% diluted with distilled water), then dyed with hematoxylin (2 g/L) for 5 min and rinsed with distilled water. The samples were then immersed in hydrochloric acid/ethanol (1 mL concentrated hydrochloric acid mixed with 99 mL 70% alcohol) for 30 s and then distilled in water for 15 min. The samples were then immersed in 1% eosin solution for 2 min and rinsed with distilled water. Finally, the tissue was dehydrated with anhydrous ethanol and sealed with a neutral resin. Partial images of the cerebral cortex of the sample were visualized under a microscope (Olympus BX51; Olympus Co., Tokyo, Japan).

### 4.7. Nissl Staining

Brain tissue sections of 10 μm were dewaxed with xylene for 10 min, then infiltrated for 10 min each with anhydrous ethanol and ethanol (95%, 75%, 60%, 50%), then subjected to double steaming and stained with thioneine (Wuhan Servicebio Technology Co. Ltd., Wuhan, China) at 37 °C for 1 h followed by differentiation with Nissl differentiation solution for several seconds. Cell morphology was observed under a microscope (Olympus BX51; Olympus Co., Tokyo, Japan)

### 4.8. Transmission Electron Microscopy (TEM)

Mitochondrial morphology was observed by transmission electron microscopy. The cerebral cortex of the infarcted side was cut into 1 mm^3^ and fixed with an electron microscope solution for 4 h. After fixation, dehydration, embedding, and cutting into 80 nm slices, tissue was stained for 15 min with 2% uranium acetate, saturated, then left in alcohol solution and lead citrate and sectioned to dry overnight at room temperature. A transmission electron microscope was used for observation and image analysis. 

### 4.9. Fe^2+^, MDA, and GSH Determination

The hemispheres of the mice were collected 24 h after reperfusion. Brain tissue was added to normal saline (9 mL/g), homogenized at 2500 g/min for 10 min on ice, and the supernatant taken as the test fluid and examined using Fe^2+^, MDA, and GSH detection kits (Jian Cheng, Nanjing, China). 

### 4.10. Extraction of Tissue Nucleoprotein

The cerebral cortex tissue of the mice was cut into small pieces and these small pieces were added to 300 μL of PBS per 50 mg, and then homogenized on ice with a homogenizer to create cell suspension. After centrifugation at 500× *g* for 3 min, cell precipitation was collected, 200 μL of plasma protein extraction reagent was added to the cell precipitation, and, after mixing, mixed well, and ice lysis was performed for 10 min. The mixture was then centrifuged at 4 °C 13,000× *g* for 10 min and the supernatant discarded. After precipitation with 600μL nuclear protein extraction reagent, mixing, ice cracking for 10 min, and centrifugation at 13,000× *g* at 4 °C for 10 min, the supernatant was nuclear protein.

### 4.11. Western Blot Analysis

After the addition of 1% PMSF lysate, cerebral cortex tissue of the ischemic side was ground into homogenate using a tissue homogenizer. After ice lysis for 30 min, samples were centrifuged at 12,000× *g* rpm and 4 °C for 30 min. Total protein content was determined using a BCA kit (Beyotime, Shanghai, China). The protein sample loading quantity in each group was controlled at 20−25 μg and separated using 10% SDS−PAGE. After target protein separation, the gel was transferred to a PVDF membrane at 80 V for 120 min. The membrane was blocked in a rapid sealing fluid at 4 °C for 30 min, then probed with the appropriate primary antibodies at 4 °C overnight: TFR1 (1:1000, Abcam, Cambridge, UK), SLC7A11 (1:1000, Proteintech, Wuhan, China), GPX4 (1:2000, Proteintech, Wuhan, China), GAPDH (1:1000, Proteintech, Wuhan, China), and Nrf2 (1:1000, CST, Danvers, MA, USA), HO−1 (1:1000, Wanlei, Shenyang, China). The membranes were then placed into TPBS and washed 3 times for 7 min each time. Next, the membranes were incubated with HRP−conjugated goat anti−rabbit IgG secondary antibodies (1:5000, Biosharp, Hefei, China) or goat anti-mouse IgG secondary antibodies (1:5000, Biosharp, Hefei, China) for 30 min and the blots detected using an ECL plus kit (Millipore, Boston, MA, USA). The membranes were visualized using a chemiluminescence detection system (VIVILBER, Paris, France) and analyzed using Image J software (Image J, Version 1.8.0, National Institutes of Health, Bethesda, MD, USA).

### 4.12. Statistical Analysis

Statistical analysis was performed with GraphPad Prism, version 7.0 (GraphPad Software, San Diego, CA, USA). Statistical significance was determined by one-way ANOVA. Data are presented as mean ± S.D. Statistical significance was set at *p* < 0.05.

## 5. Conclusions

Our study demonstrated that PCs alleviated CIRI by inhibiting ferroptosis via the Nrf2/HO−1 signaling pathway. We provided scientific evidence that PCs can protect against CIRI and may serve as a functional food.

## Figures and Tables

**Figure 1 molecules-28-03582-f001:**
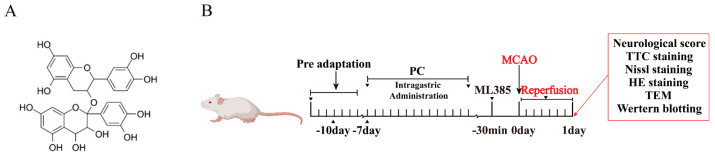
Chemical structure of procyanidins (**A**) and the experimental plan (**B**).

**Figure 2 molecules-28-03582-f002:**
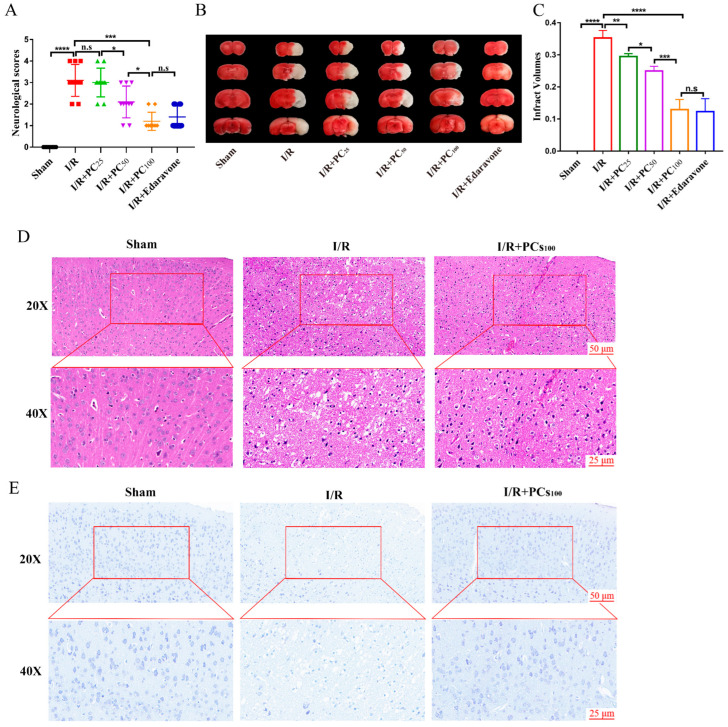
Effects of procyanidins on the neurological deficit, cerebral infarct volume, and pathological lesions following MCAO/R in mice. (**A**) Neurological scores of mice with CIRI (n = 10). (**B**,**C**) The cerebral infarct volume in mice after CIRI (n = 6). (**D**) Representative HE-stained sections of mouse cortex. (**E**) Representative Nissl−-stained sections of mouse cortex. n.s: no significant difference, * *p* < 0.05, ** *p* < 0.01, *** *p* < 0.001, **** *p* < 0.0001.

**Figure 3 molecules-28-03582-f003:**
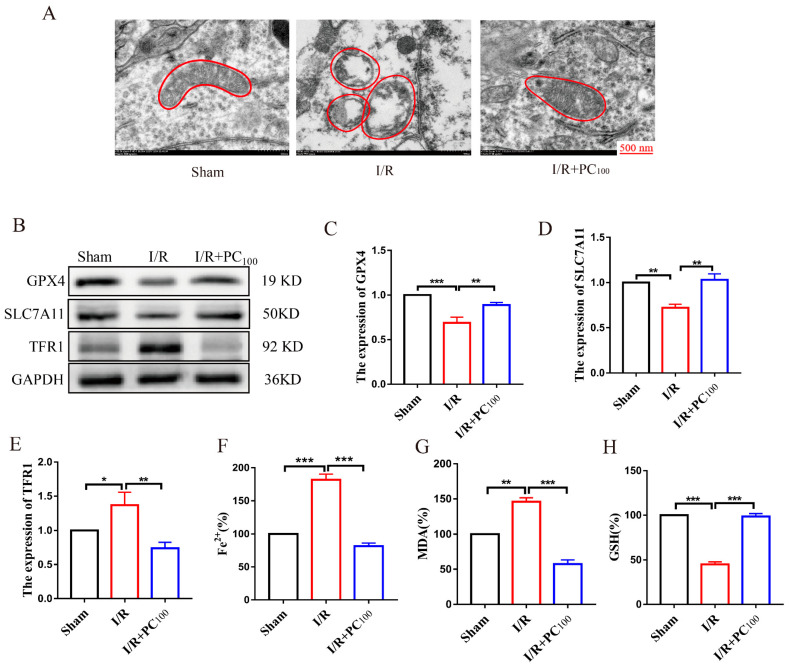
Effects of procyanidin inhibition on cerebral I/R−induced ferroptosis. (**A**) The effect of PC treatment on mitochondrial morphology in mice with CIRI. (**B**–**E**) The expression of GPX4, SLC7A11, and TFR1 in the brain tissue of mice with CIRI (n = 3). (**F**,**G**) Level of Fe^2+^ and MDA in the brain tissue of mice with CIRI (n = 3). (**H**) Level of GSH in the brain tissue of mice with CIRI (n = 3). All values are expressed as mean ± SD. n.s: no significant difference, * *p* < 0.05, ** *p* < 0.01, *** *p* < 0.001.

**Figure 4 molecules-28-03582-f004:**
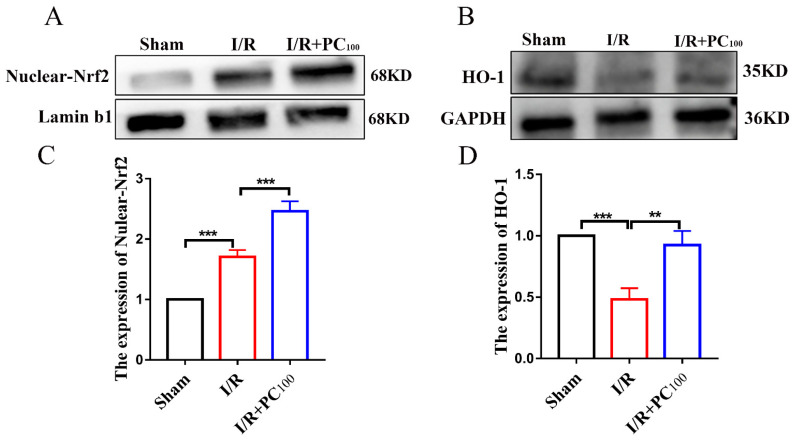
Effects of procyanidins on the Nrf2/HO-1 pathway. (**A**−**D**) The expression of Nuclear-Nrf2 and HO−1 in the brain tissue of mice with CIRI (n = 3). All values are expressed as mean ± SD. n.s: no significant difference, ** *p* < 0.01, *** *p* < 0.001.

**Figure 5 molecules-28-03582-f005:**
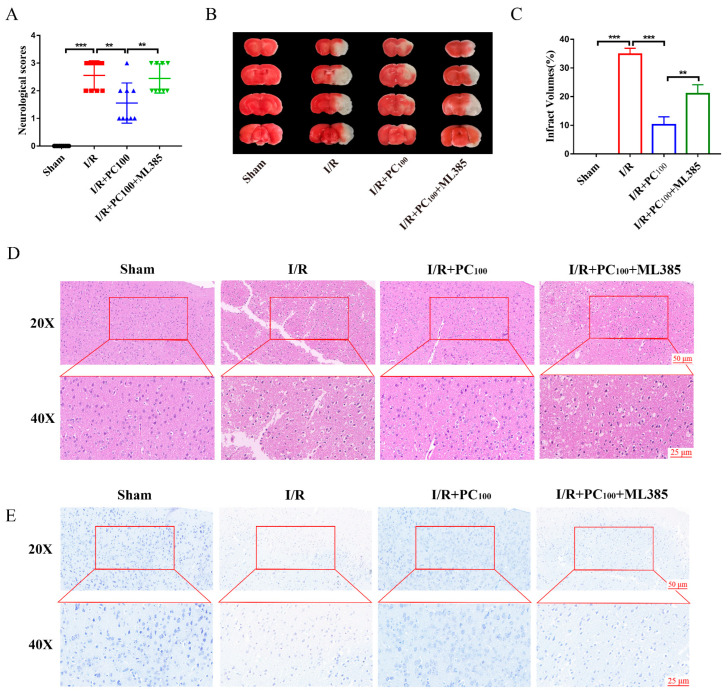
ML385 reversed the neuroprotective effect of procyanidins. (**A**) Neurological scores of mice with CIRI. (**B**,**C**) The cerebral infarct volume in mice after CIRI (n = 6). (**D**) Representative HE−stained sections of the mouse cortex. (**E**) Representative Nissl-stained sections of the mouse cortex. All values are expressed as mean ± SD. n.s: no significant difference, ** *p* < 0.01, *** *p* < 0.001.

**Figure 6 molecules-28-03582-f006:**
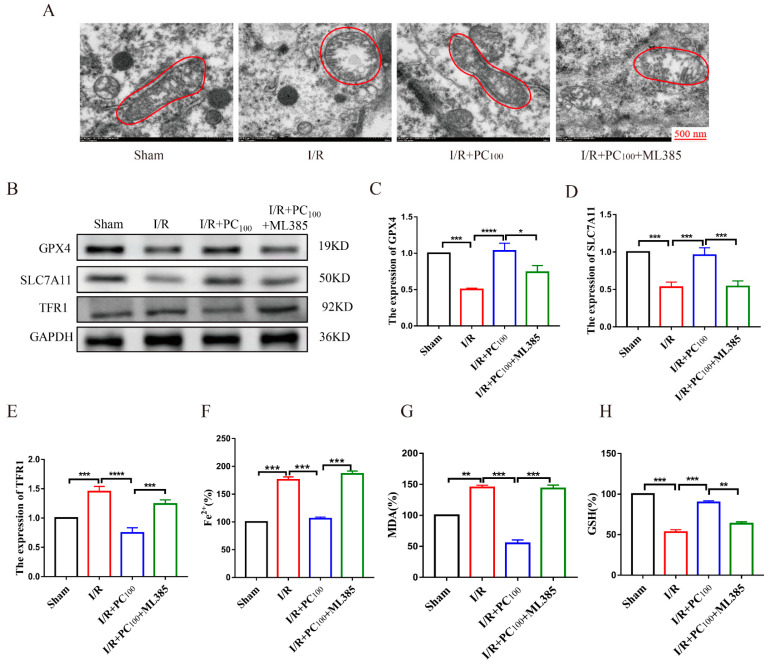
ML385 reversed the anti-ferroptotic effects of procyanidins in mice after CIRI. (**A**) Effect of ML385 on mitochondrial morphology in mice with CIRI. (**B**−**E**) Effect of ML385 on the expression of GPX4, SLC7A11, and TFR1 (n = 3). (**F**,**G**) Effect of ML385 on Fe^2+^ and MDA in mice with CIRI (n = 3). (**H**) Level of GSH in the brain tissue of mice with CIRI (n = 3). All values are expressed as mean ± SD. n.s: no significant difference, * *p* < 0.05, ** *p* < 0.01, *** *p* < 0.001, **** *p* < 0.0001.

**Figure 7 molecules-28-03582-f007:**
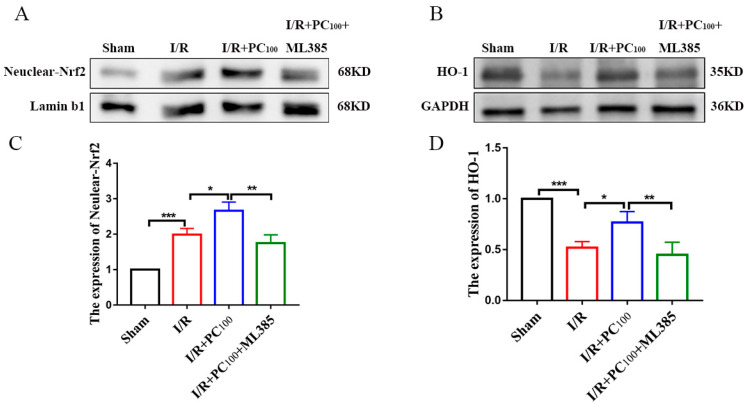
Procyanidins activated the Nrf2/HO−1 pathway. (**A**–**D**) Expression of Nuclear−Nrf2 and HO−1 in the brain tissue of mice with CIRI (n = 3). All values are expressed as mean ± SD. n.s: no significant difference, * *p* < 0.05, ** *p* < 0.01, *** *p* < 0.001.

**Figure 8 molecules-28-03582-f008:**
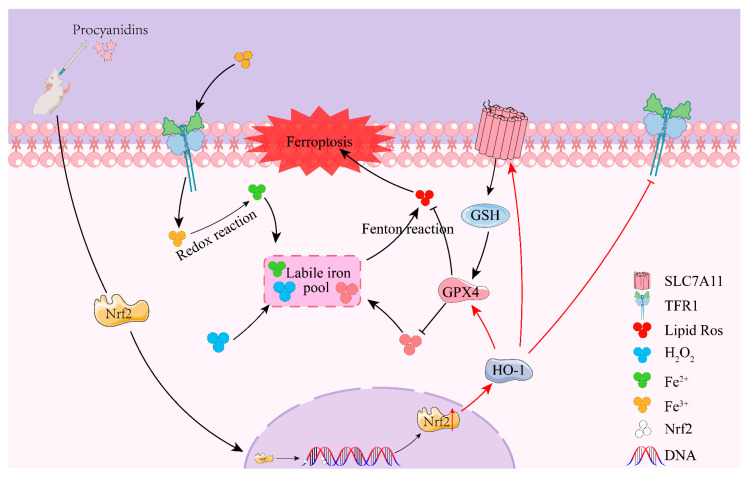
Schematic representation of the proposed neuroprotective mechanism of procyanidins following CIRI.

## Data Availability

Not applicable.

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
