# Peer review of "Procyanidins Alleviated Cerebral Ischemia/Reperfusion Injury by Inhibiting Ferroptosis via the Nrf2/HO-1 Signaling Pathway"

_molecules, 2023, doi:10.3390/molecules28083582_

Round 1
Reviewer 1 Report
There is good data to support a scientific discussion in the manuscript. I hope readers will enjoy reading this article. This is an interesting study in which methods and experiments are documented in detail. The results are presented appropriately, clearly explaining the study results. The manuscript is written in a direct and active style. Overall, the paper offers sufficient detail on the methodology and is written comprehensively enough to be understandable.
1. Abstract: Authors must rewrite the abstract and organize the information in such a way as to make each part of an abstract clear. Introduction, hypothesis, and the objective of the work, not to delve so deeply into the experimental design and methodology, but to make it clear that they measured and how they measured it, briefly results and the conclusion of the work.
2. What is the purpose of adding I/R + Edaravone (30mg/kg) to this group? I did not find a discussion concerning this addition
3. I advise the authors to broaden the discussions, in light of the interesting results obtained in the experimental section.
4. The authors should emphasize the importance of mitochondrial morphology changes following IR or PC treatment in their Discussion.
5. It appears that the authors have expressed nuclear NRF2, but I did not find how to isolate nuclear proteins in their methods.
6. Histopathological evaluation of brain tissue_ This section of results must be correctly described with an adequate scientific vocabulary. The images are not clear. And nowhere do they mention that the areas they evaluate are the cortex, until they show the figure. Mention of the explored area should be made from the methods section.
Author Response
Dear reviewer:
Thank you very much for your comments and professional advice.This opinions help to improve academic rigor of our article. Based on your suggestion and request, we have made corrected modifications on the revised manuscript. Meanwhile,we would like to show the details as follows:
Reviewer 1#
- Abstract: Authors must rewrite the abstract and organize the information in such a way as to make each part of an abstract clear. Introduction, hypothesis, and the objective of the work, not to delve so deeply into the experimental design and methodology, but to make it clear that they measured and how they measured it, briefly results and the conclusion of the work.
The author’s answer: We have re-written this part according to reviewer’s suggestion. Please refer to lines 11-24.
- What is the purpose of adding I/R + Edaravone to this group? I did not find a discussion concerning this addition
The author’s answer: I/R + Edaravone (30mg/kg) was used as a positive control group to illustrate the effectiveness of PCs. We have added related descriptions in the discussion section. Please refer to lines 263-265.
- I advise the authors to broaden the discussions, in light of the interesting results obtained in the experimental section.
The author’s answer: Thank you very much for your advice. We've expanded on the interesting parts, lines 283 to 291.
- The authors should emphasize the importance of mitochondrial morphology changes following IR or PC treatment in their Discussion.
The author’s answer: Thank you very much for your comments. We have emphasized the importance of mitochondrial morphological changes in our discussion, which is referred to in lines 263-266.
- It appears that the authors have expressed nuclear NRF2, but I did not find how to isolate nuclear proteins in their methods.
The author’s answer: Thank you very much for your suggestion. For the description of nucleoprotein extraction, we have added lines 401-410 to the manuscript for reference.
- Histopathological evaluation of brain tissue This section of results must be correctly described with an adequate scientific vocabulary. The images are not clear. And nowhere do they mention that the areas they evaluate are the cortex, until they show the figure. Mention of the explored area should be made from the methods section.
The author’s answer: We think it is excellent suggestion. We conducted a scientific HE description, including the exact location where the change can be found in the revised manuscript (line 95-99). In the newly uploaded manuscript we have emphasized in the method section that the image is taken from the cerebral cortex. At the same time, HE staining results were redescribed. In addition, we have uploaded higher resolution images of HE staining (line 105,161).
We tried our best to improve the manuscript and made some changes marked in red in revised paper which will not influence the content and framework of the paper. Once again, thank you very much for your comments and suggestions.
Yours sincerely,
Lei Chen
Corresponding author: Shu-ying Dong
Email: 0900007@bbmc.edu.cn
Reviewer 2 Report
In this manuscript authors were trying to demonstrate that Procyanidins (PC), a natural antioxidants with variety of functions, such as oxidative stress inhibition, anti-apoptotic, and metal ions chelation was neuroprotective after cerebral ischemia/reperfusion injury. Mechanism study further suggested that PC performed its neuroprotective role might via increasing the expression of Nuclear-Nrf2 and HO-1 to inhibit ferroptosis. While the manuscript was fairly clearly written and its major points are sound, there are several issues that need to be fixed that would significantly bolster its clinic relevance.
Major:
1. Pre-injury long term treatment is not a practical intervention for using PC to treat cerebral ischemia/reperfusion injury in clinic field. Authors should try and demonstrate that PC post-injury treatment is neuroprotective as well.
2. Lack of chronic functional outcome evaluation further mitigated the impact of using PC to treat cerebral. ischemia/reperfusion injury regarding the neuroprotective effects.
3. Regarding its function of metal ions chelator, the iron level after PC treatment should be evaluated or at least be discussed.
Minors:
1. There are some writing issues. For example, in line 42 and 43: “Studies have shown that PCs can alleviate cerebral I/R injury, but few studies have examined the protective effect of PCs on cerebral I/R injury.” What is logical between these two sentences? If studies have shown PCs can alleviate cerebral I/R injury, it means PCs are protective. I guess authors are more like to say: but few studies have examined the mechanisms underlie the protective effect of PCs on cerebral I/R injury.
2. All images in Figures have no scale bar.
3. Image quality, especially the high power ones is not high enough to clearly see single cell structure.
4. Some Figure labeling is unclear. For example, in Figure 2, c, what is unit for infarct volume?
Author Response
Dear reviewer:
Thank you very much for your comments and professional advice.This opinions help to improve academic rigor of our article. Based on your suggestion and request, we have made corrected modifications on the revised manuscript. Meanwhile,we would like to show the details as follows:
Reviewer 2#
Major:
- Pre-injury long term treatment is not a practical intervention for using PC to treat cerebral ischemia/reperfusion injury in clinic field. Authors should try and demonstrate that PC post-injury treatment is neuroprotective as well.
The author’s answer: Thank you very much for your the comment. In this paper, we aim to explore the protective effects of PCs on cerebral I/R injury. Firstly, cerabral I/R injury is an acute pathological process, which can cause serious brain injury in the early stage of reperfusion, while it takes time for PCs to exert protective effects. Secondly, PCs exists in many plants and fruits and is a natural metal ion chelating agent and antioxidant. The protecive effects of PCs preadministration indicate that functional foods which are rich in PCs may benefit people who are likely to suffer a stroke. Furthermore, patients with ischemic stroke are prone to suffer a second stroke. Just as the the comment, if the neuroprotective effects of PC post-injury treatment is demonstrated, it will be more practical, which provides ideas for our future research.
- Lack of chronic functional outcome evaluation further mitigated the impact of using PC to treat cerebral ischemia/reperfusion injury regarding the neuroprotective effects.
The author’s answer: Thanks a lot for your comment. Ischemia/reperfusion injury is an acute pathological process, especially in the process of blood reperfusion, which can cause serious cascade reaction. Early preventive intervention alleviated ferroptosis induced by lipid peroxidation. In this way, we confirmed that PCs alleviated cerebral I/R injury in mice. However, it is still unclear whether PCs has a role in slow functional recovery in mice, which should be evaluated in the the study of protective effects of PCs post-injury treatment.
- Regarding its function of metal ions chelator, the iron level after PC treatment should be evaluated or at least be discussed.
The author’s answer: Thank you very much for your suggestion. We have discussed the related description of the effect of PCs administration on iron levels (Line 268-277).
Minors:
- There are some writing issues. For example, in line 42 and 43: “Studies have shown that PCs can alleviate cerebral I/R injury, but few studies have examined the protective effect of PCs on cerebral I/R injury.” What is logical between these two sentences? If studies have shown PCs can alleviate cerebral I/R injury, it means PCs are protective. I guess authors are more like to say: but few studies have examined the mechanisms underlie the protective effect of PCs on cerebral I/R injury.
The author’s answer: Thank you very much for your attention to the details of this manuscript, which will help us to publish a better paper. We have corrected the inappropriate statements in lines 42 and 43(Line 57-58).
- All images in Figures have no scale bar.
The author’s answer: We've added a scale bar to all the images (Line 105, 132, 161, 181).
- Image quality, especially the high power ones is not high enough to clearly see single cell structure.
The author’s answer: The low resolution image have been replaced (Line 105 and 161).
- Some Figure labeling is unclear. For example, in Figure 2, c, what is unit for infarct volume?
The author’s answer: The unit of cerebral infarction volume was %, and we added units to all relevant images (Line 105 and 161).
We tried our best to improve the manuscript and made some changes marked in red in revised paper which will not influence the content and framework of the paper. Once again,thank you very much for your comments and suggestions.
Yours sincerely,
Lei Chen
Corresponding author: Shu-ying Dong
Email: 0900007@bbmc.edu.cn
Reviewer 3 Report
The article, Procyanidins alleviated cerebral ischemia/reperfusion injury by inhibiting ferroptosis via the Nrf2/HO-1 signalling pathway, accept in present form
Author Response
Thank you for your appreciation on the manuscript.
Round 2
Reviewer 1 Report
Satisfy with author comments
Reviewer 2 Report
Manuscript needs to be carefully read to avoid some misspelling and typo.